# ADVERSARIALLY ROBUST TRANSFER LEARNING

**Ali Shafahi**[*][†]**, Parsa Saadatpanah**[*][†]**, Chen Zhu**[*][†]**, Amin Ghiasi**[†]**, Cristoph Studer**[‡]**,**
{ashafahi,parsa,chenzhu,amin}@cs.umd.edu; studer@cornell.edu
**David Jacobs**[†]**, Tom Goldstein**[†]
{djacobs,tomg}@cs.umd.edu

## ABSTRACT

Transfer learning, in which a network is trained on one task and re-purposed on another, is often used to produce neural network classifiers when data is scarce or full-scale training is too costly. When the goal is to produce a model that is not only accurate but also adversarially robust, data scarcity and computational limitations become even more cumbersome. We consider robust transfer learning, in which we transfer not only performance but also robustness from a source model to a target domain. We start by observing that robust networks contain robust feature extractors. By training classifiers on top of these feature extractors, we produce new models that inherit the robustness of their parent networks. We then consider the case of "fine tuning" a network by re-training end-to-end in the target domain. When using lifelong learning strategies, this process preserves the robustness of the source network while achieving high accuracy. By using such strategies, it is possible to produce accurate and robust models with little data, and without the cost of adversarial training. Additionally, we can improve the generalization of adversarially trained models, while maintaining their robustness.

## 1 INTRODUCTION

Deep neural networks achieve human-like accuracy on a range of tasks when sufficient training data and computing power is available. However, when large datasets are unavailable for training, or pracitioners require a low-cost training strategy, *transfer learning* methods are often used. This process starts with a source network (pre-trained on a task for which large datasets are available), which is then re-purposed to act on the target problem, usually with minimal re-training on a small dataset (Yosinski et al., 2014; Pan & Yang, 2009).

While transfer learning greatly accelerates the training pipeline and reduces data requirements in the target domain, it does not address the important issue of model *robustness*. It is well-known that naturally trained models often completely fail under adversarial inputs (Biggio et al., 2013; Szegedy et al., 2013). As a result, researchers and practitioners often resort to *adversarial training*, in which adversarial examples are crafted on-the-fly during network training and injected into the training set. This process greatly exacerbates the problems that transfer learning seeks to avoid. The high cost of creating adversarial examples increases training time (often by an order of magnitude or more). Furthermore, robustness is known to suffer when training on a small dataset (Schmidt et al., 2018). To make things worse, high-capacity models are often needed to achieve good robustness (Madry et al., 2017; Kurakin et al., 2016; Shafahi et al., 2019b), but these models may over-fit badly on small datasets.

### CONTRIBUTIONS

The purpose of this paper is to study the adversarial robustness of models produced by transfer learning. We begin by observing that robust networks contain *robust feature extractors*, which are resistant to adversarial perturbations in different domains. Such robust features can be used

---

[*]equal contribution
[†]University of Maryland
[‡]Cornell University

as a basis for semi-supervised transfer learning, which only requires re-training the last layer of a network. To demonstrate the power of robust transfer learning, we transfer a robust ImageNet source model onto the CIFAR domain, achieving both high accuracy and robustness in the new domain without adversarial training. We use visualization methods to explore properties of robust feature extractors. Then, we consider the case of transfer of learning by "fine-tuning." In this case, the source network is re-trained end-to-end using a small number of epochs on the target domain. Unfortunately, this end-to-end process does not always retain the robustness of the source domain; the network "forgets" the robust feature representations learned on the source task. To address this problem, we use recently proposed lifelong learning methods that prevent the network from forgetting the robustness it once learned. Using our proposed methods, we construct robust models that generalize well. In particular, we improve the generalization of a robust CIFAR-100 model by roughly 2% while preserving its robustness.

## 2 BACKGROUND

Adversarial examples fall within the category of evasion attacks—test-time attacks in which a perturbation is added to a natural image before inference. Adversarial attacks are most often crafted using a differentiable loss function that measures the performance of a classifier on a chosen image. In the case of norm-constrained attacks (which form the basis of most standard benchmark problems), the adversary solves

$$\max_{\delta} \quad l(x + \delta, y, \theta) \qquad \text{s.t.} \quad \|\delta\|_p \le \epsilon, \tag{1}$$

where $\theta$ are the (already trained and frozen) parameters of classifier $c(x, \theta) \to \hat{y}$ that maps an image to a class, $l$ is the proxy loss used for classification (often cross-entropy), $\delta$ is the image perturbation, $(x, y)$ is the natural image and its true class, and $\|.\|_p$ is some $\ell_p$-norm[1]. The optimization problem in Eq. 1 aims to find a bounded perturbation that maximizes the cross-entropy loss given the correct label. There are many variants of this process, including DeepFool (Moosavi-Dezfooli et al., 2016), L-BFGS (Szegedy et al., 2013), and CW (Carlini & Wagner, 2017).

Many researchers have studied methods for building a robust network which have been later shown to be ineffective when attacked with stronger adversaries (Athalye et al., 2018). Adversarial training (Szegedy et al., 2013) is one of the defenses that was not broken by Athalye et al. (2018). While adversarial training using a weak adversary such as the FGSM attack (Goodfellow et al., 2015) can be broken even by single step attacks which add a simple random step prior to the FGSM step (Tramèr et al., 2017), adversarial training using a strong attack has successfully improved robustness. Madry et al. (2017) showed that a PGD attack (which is a BIM attack (Kurakin et al., 2016) with an initial random step and projection) is a strong enough attack to achieve promising adversarial training results. We will refer to this training method as *PGD adversarial training*. PGD adversarial training achieves good robustness on bounded attacks for MNIST (LeCun et al., 1998) and acceptable robustness on CIFAR-10 (Krizhevsky & Hinton, 2009) classifiers.

Tsipras et al. (2018) show that adversarial training with strong PGD adversaries has many benefits in addition to robustness. They also state that while adversarial training may improve generalization in regimes where training data is limited (especially on MNIST), it may be at odds with generalization in regimes where data is available. This trade-off was also recently studied by Zhang et al. (2019), Su et al. (2018), and Shafahi et al. (2019a).

While, to the best of our knowlegde, the transferability of robustness has not been studied in depth, Hendrycks et al. (2019) studied the case of adversarially training models that were pre-trained on different domains. Our work is fundamentally different in that we seek to transfer robustness without resorting to costly and data-hungry adversarial training. We train the target model on natural examples only, which allows us to directly study how well robustness transfers. Additionally, this allows us to have better generalization and achieve higher accuracy on validation examples. While as Hendrycks et al. (2019) state, fine-tuning on adversarial examples built for the target domain can improve robustness of relatively large datasets such as CIFAR-10 and CIFAR-100 compared to adversarial training from scratch on the target domain, we show that in the regimes of limited data (where transfer learning is more common), adversarially robust transfer learning can lead to better results measured in terms of both robustness and clean validation accuracy.

---

[1]By default we will use the $\ell_\infty$-norm in this paper.

Table 1: Accuracy and robustness of natural and adversarially trained models on CIFAR-10+ and CIFAR-100+. The "+" sign denotes standard data augmentation ($\epsilon = 8$).

| Dataset | model | validation accuracy | accuracy on PGD-20 | accuracy on CW-20 |
|---------|-------|---------------------|--------------------|--------------------|
| CIFAR-10+ | natural | 95.01% | 0.00% | 0.00% |
|  | robust | 87.25% | 45.84% | 46.96% |
| CIFAR-100+ | natural | 78.84% | 0.00% | 0.00% |
|  | robust | 59.87% | 22.76% | 23.16% |

## 3 THE ROBUSTNESS OF DEEP FEATURES

In this section, we explore the robustness of different network layers, and demonstrate that robust networks rely on robust deep features. To do so, we start from robust classifiers ($c(\theta_r)$) for the CIFAR-100 and CIFAR-10 datasets (Krizhevsky & Hinton, 2009), and update $\theta$ by training on natural examples. In each experiment, we re-initialize the last $k$ layers/blocks of the network, and re-train just those layers. We start by re-initializing just the last layer, then the last two, and so on until we re-initialize all the layers.

We use the adversarially trained Wide-ResNet 32-10 (Zagoruyko & Komodakis, 2016) for CIFAR-10 from Madry et al. (2017) as our robust model for CIFAR-10. We also adversarially train our own robust classifier for CIFAR-100 using the code from Madry et al. (2017). To keep things consistent, we use the same hyper-parameters used by Madry et al. (2017) for adversarially training CIFAR-10 to adversarially train the CIFAR-100 model.[2] The performance of the CIFAR-10 and CIFAR-100 models on natural and adversarial examples are summarized in Table 1. To measure robustness, we evaluate the models on adversarial examples built using PGD attacks.

We break the WRN 32-10 model into 17 blocks, which are depicted in Fig. 2. In each experiment, we first re-initialize the $k$ deepest blocks (blocks 1 through $k$) and then train the parameters of those blocks on natural images[3]. We train for 20,000 iterations using Momentum SGD and a learning rate of 0.001. We then incrementally unfreeze and train more blocks. For each experiment, we evaluate the newly trained model's accuracy on validation adversarial examples built with a 20-step PGD $\ell_\infty$ attack with $\epsilon = 8$.

Fig. 1 shows that robustness does not drop if only the final layers of the networks are re-trained on natural examples. In fact, there is a slight increase in robustness compared to the baseline PGD-7 adversarially trained models when we just retrain the last batch-normalization block and fully connected block. As we unfreeze and train more blocks, the network's robustness suddenly drops. This leads us to believe that a hardened network's robustness is mainly due to robust deep feature representations and robustness is preserved if we re-train on top of deep features.

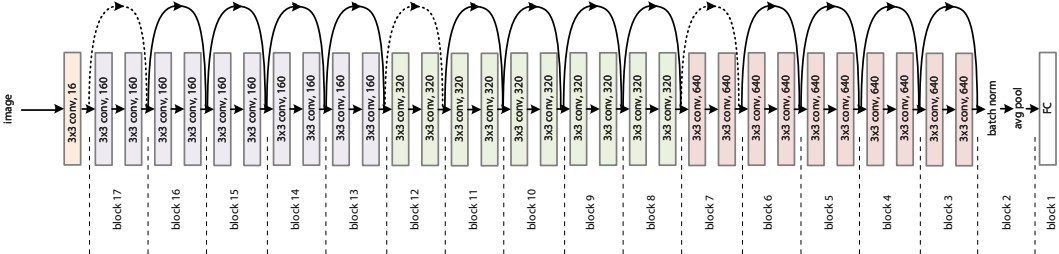

Figure 2: Wide Resnet 32-10 and the blocks used for freezing/retraining

Now that we have identified feature extractors as a source of robustness, it is natural to investigate whether robustness is preserved when transfer learning using robust feature extractors. We will

---

[2]We adv. train the WRN 32-10 on CIFAR-100 using a 7-step $\ell_\infty$ PGD attack with step-size=2 and $\epsilon = 8$. We train for 80,000 iterations with a batch-size of 128.

[3]In this experiment, we use standard data augmentation techniques.

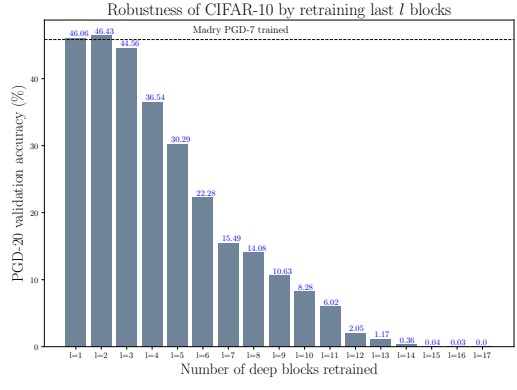
(a) CIFAR-10 PGD-20 accuracy

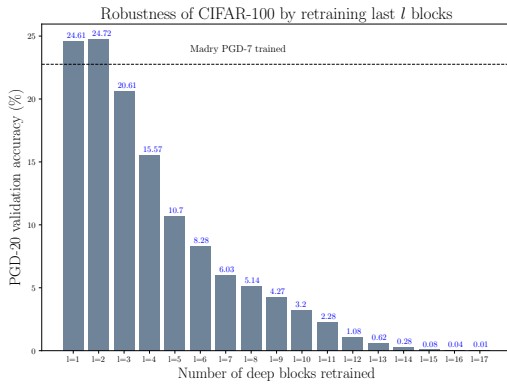
(b) CIFAR-100 PGD-20 accuracy

Figure 1: Robustness is preserved when we retrain only the deepest block(s) of robust CIFAR-10 and CIFAR-100 models using natural examples. The vertical axis is the accuracy on PGD-20 generated adversarial examples (*i.e.* robustness) after re-training deep layers. The robustness of the adversarially trained models if all layers are frozen are shown with dashed lines.

study two different approaches for transferring robustness across datasets: one in which only the last layer is re-trained, and one with end-to-end re-training.

## 4  TRANSFER LEARNING: RECYCLING FEATURE EXTRACTORS

We study how robustness transfers when the feature extractor layers of the source network are frozen, and we retrain only the last fully connected layer (*i.e.* the classification layer) for the new task. Formally, the transfer learning objective is:

$$\min_{w} \quad l(z(x, \theta^*), y, w) \tag{2}$$

where $z$ is the deep feature extractor function with pre-trained and now "frozen" parameters $\theta^*$, and $w$ represents the trainable parameters of the last fully connected layer. To investigate how well robustness transfers, we use two source models: one that is hardened by adversarial training and another that is naturally trained.

We use models trained on CIFAR-100 as source models and perform transfer learning from CIFAR-100 to CIFAR-10. The results are summarized in Table 2. Compared to adversarial/natural training the target model, transferring from a source model seems to result in a drop in natural accuracy (compare first row of Table 1 to the first row of Table 2). This difference is wider when the source and target data distributions are dissimilar (Yosinski et al., 2014).

To evaluate our method on two datasets with more similar attributes, we randomly partition CIFAR-100 into two disjoint subsets where each subset contains images corresponding to 50 classes. Table 2 shows the accuracy of transferring from one of the disjoint sets to the other (second row) and to the same set (third row). We can compare results of transfer learning with adversarial training on CIFAR-100 by averaging the results in the second and third rows of Table 2 to get the accuracy across all 100 classes of CIFAR-100.[4] By doing so, we see that the accuracy of the transferred classifier matches that of the adversarially trained one, even though no adversarial training took place in the target domain. For completeness, we have also included experiments where we use CIFAR-10 as the source and CIFAR-100 as the target domain.

We make the following observations from the transfer-learning results in Table 2. 1) *robustness transfers:* when the source model used for transfer learning is robust, the target model is also robust (although less so than the source), 2) *robustness transfers between models that are more similar:* If

---

[4]The robust CIFAR-100 classifier has 59.87% validation accuracy and 22.76% accuracy on PGD-20 adversarial examples. The average validation accuracy of the two half-CIFAR-100 classifiers on validation examples is $\frac{64.96\% + 58.48\%}{2} = 61.72\%$ while the average robustness is $\frac{25.16\% + 15.86\%}{2} = 20.51\%$.

Table 2: Transfer learning by freezing the feature extractor layers ($\epsilon = 8$).

| Source Dataset | Target Dataset | Source Model | val. | PGD-20 | CW-20 |
|---|---|---|---|---|---|
| CIFAR-100 | CIFAR-10 | natural | 83.05% | 0.00% | 0.00% |
| | | robust | 72.05% | 17.70% | 17.43% |
| CIFAR-100 (50% of classes) | CIFAR-100 (other 50% of classes) | natural | 71.44% | 0.00% | 0.00% |
| | | robust | 58.48% | 15.86% | 15.30% |
| CIFAR-100 (50% of classes) | CIFAR-100 (same 50% of classes) | natural | 80.20% | 0.00% | 0.00% |
| | | robust | 64.96% | 25.16% | 25.56% |
| CIFAR-10 | CIFAR-100 | natural | 49.66% | 0.00% | 0.00% |
| | | robust | 41.59% | 11.63% | 9.68% |

Table 3: Transfer learning from ImageNet ($\epsilon = 5$).

| Architecture and Source Dataset | Target Dataset | Source Model | val. | PGD-20 | CW-20 |
|---|---|---|---|---|---|
| ResNet-50 ImageNet | CIFAR-10+ | natural | 90.49% | 0.01% | 0.00% |
| | | robust ($\epsilon = 5$) | 88.33% | 22.66% | 26.01% |
| | CIFAR-100+ | natural | 72.84% | 0.05% | 0.00% |
| | | robust ($\epsilon = 5$) | 68.88% | 15.21% | 18.34% |
| Robust u-ResNet-50 ($\epsilon = 5$) for CIFAR-10+ | | | 82.00% | 53.11% | |
| Robust u-ResNet-50 ($\epsilon = 5$) for CIFAR-100+ | | | 59.90% | 29.54% | |

the source and target models are trained on datasets which have similar distributions (and number of classes), robustness transfers better, and 3) *validation accuracy is worst if we use a robust model as the source compared to using a conventionally trained source model:* if the source model is naturally trained, the natural validation accuracy is better, although the target model is then vulnerable to adversarial perturbations.

## 4.1 TRANSFER LEARNING WITH IMAGENET MODELS

Transfer learning using models trained on ImageNet (Russakovsky et al., 2015) as the source is a common practice in industry because ImageNet feature extractors are powerful and expressive. In this section we evaluate how well robustness transfers from these models.

### 4.1.1 TRANSFER LEARNING USING IMAGENET

Starting from both a natural and robust ImageNet model, we perform the same set of experiments we did in section 4. Robust ImageNet models do not withstand untargeted $\ell_\infty$ attacks using as large an $\epsilon$ as those that can be used for simpler datasets like CIFAR. Following the method Shafahi et al. (2019b), we "free train" a robust ResNet-50 on ImageNet using replay hyper-parameter $m = 4$. The hardened ImageNet classifier withstands attacks bounded by $\epsilon = 5$. Our robust ImageNet achieves 59.05% top-1 accuracy and roughly 27% accuracy against PGD-20 $\ell_\infty$ $\epsilon = 5$ attacks on validation examples. We experiment with using this robust ImageNet model and a conventionally trained ResNet-50 ImageNet model as the source models.

Using the ImageNet source models, we train CIFAR classifiers by retraining the last layer on natural CIFAR examples. We up-sample the $32 \times 32$-dimensional CIFAR images to $224 \times 224$ before feeding them into the ResNet-50 source models that are trained on ImageNet. For evaluation purposes, we also train robust ResNet-50 models from scratch using (Shafahi et al., 2019b) for the CIFAR models. To ensure that the transfer learning models and the end-to-end trained robust models have the same capacity and dimensionality, we first upsample the CIFAR images before feeding them to the ResNet-50 model. To distinguish between the common case of training ResNet models on CIFAR images that are $32 \times 32$-dimensional, we call our models that are trained on the upsampled CIFAR datasets the upsample-first ResNets or "*u-ResNets*".

Table 3 illustrates that using a robust ImageNet model as a source results in high validation accuracy for the transferred CIFAR target models. Also, given that the ImageNet classifier by itself is 27% robust, the CIFAR-10 model maintains the majority of that 27% robustness. When we compare the end-to-end hardened classifiers (robust u-ResNets) with the transferred classifier, we can see that

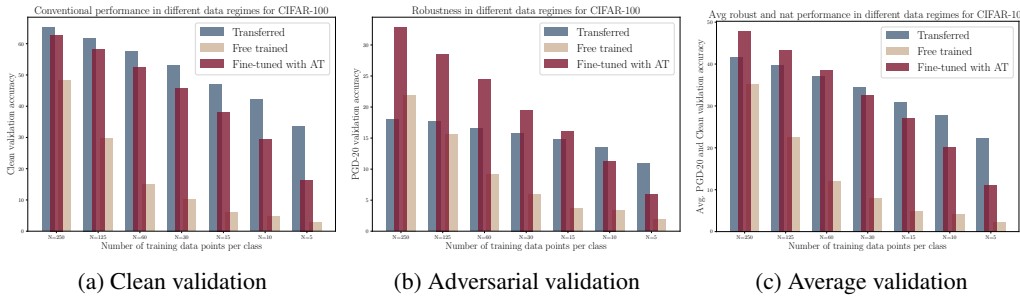

(a) Clean validation  (b) Adversarial validation  (c) Average validation

Figure 3: When the number of training data per-class is very limited (right bars), adversarially robust transfer learning [Transferred] is better in all metrics. However, as the number of training data increases (left bars), fine-tuning with adversarial examples of the target domain [Fine-tuned with AT] results in more robustness. Adversarially robust transfer learning always results in models that work better on natural examples and is $3\times$ faster than fine-tuning with adversarial examples of the target domain. Using a pre-trained robust ImageNet improves both robustness and generalization.

while the robustness is less for the transferred case, transferred models result in considerably better performance on clean validation examples.

## 4.2 LOW-DATA REGIME

As touched on before, transfer learning is more common in situations where the number of training points in the target domain is limited. Up until now, as a proof of concept, we have illustrated the majority of our experiments on the CIFAR target domains where we have many training points per-class. Hendrycks et al. (2019) show that starting from a pre-trained robust ImageNet model and fine-tuning on adversarial examples of the CIFAR domain can improve robustness beyond that of simply adversarial training CIFAR. Here, we illustrate the effect of training data size on robustness and natural performance by running various experiments on subsets of CIFAR-100 where we vary the number of training points per-class ($N$).

We compare three different hardening methods: (1) Free-training/adversarial training the target domain (Shafahi et al., 2019b); (2) fine-tuning using adversarial examples of the target task starting from the Free-4 robust ImageNet model similar to (Hendrycks et al., 2019); and (3) training a fully connected layer on top of the frozen feature extractors of the Free-4 robust ImageNet model using natural examples from the target task. For comparing the three different approaches, we look at three metrics: (a) clean validation accuracy; (b) robustness against PGD-20 validation adversarial examples; and (c) average of robustness and clean performance (((a)+(b))/2.) The results are summarized in Fig. 3. In the regimes where transfer learning is more common, adversarially robust transfer learning results in the best overall performance. Adversarially/Free training the target domain results in less robustness and validation accuracy compared to fine-tuning which highlights the importance of pre-training (Hendrycks et al., 2019). Note that in terms of computational resources required, the cost of fine-tuning on adversarial examples of the target domain is about $k\times$ our method since it requires generation of adversarial examples using $k$-step PGD attacks (we set $k = 3$).

### 4.2.1 TRAINING DEEPER NETWORKS ON TOP OF ROBUST FEATURE EXTRACTORS

The basic transfer learning setting of section 4.1.1 only re-trains one layer for the new task. In section 4.1.1, when we transferred from the robust ImageNet to CIFAR-100, the natural *training* accuracy was 88.84%. Given the small number of trainable parameters left for the network ($\approx 2048 \times 100$) and the fixed feature extractor, the network was not capable of completely fitting the training data. This means that there is potential to improve natural accuracy by learning more complex non-linear features and increasing the number of trainable parameters.

To increase representation capacity and the number of trainable parameters, instead of training a 1-layer network on top of the feature extractor, we train a multi-layer perceptron (MLP) network on top of the robust feature extractor. To keep things simple and prevent bottle-necking, every hidden layer we add has 2048 neurons. We plot the training and validation accuracies on the natural examples and

the robustness (*i.e.* PGD-20 validation accuracy) in Fig. 4 for various numbers of hidden layers. As can be seen, adding one layer is enough to achieve 100% training accuracy. However, doing so does not result in an increase in validation accuracy. To the contrary, adding more layers can result in a slight drop in validation accuracy due to overfitting. As illustrated, we can improve generalization using simple but effective methods such as dropout (Srivastava et al., 2014) (with probability 0.25) and batch-normalization (Ioffe & Szegedy, 2015).

However, the most interesting behavior we observe in this experiment is that, as we increase the number of hidden layers, the robustness to PGD-20 attacks improves. Note, this seems to happen even when we transfer from a naturally trained ImageNet model. While for the case where we have no hidden layers, robustness is 0.00% on CIFAR100 when we use a naturally trained ImageNet model as source, if our MLP has 1, 2, 3, or 5 hidden layers, our robustness against PGD attacks would be 0.03%, 0.09%, 0.31% and 6.61%, respectively. This leads us to suspect that this behavior may be an artifact of vanishing gradients for adversary as the softmax loss saturates when the data is fit perfectly (Athalye et al., 2018). Therefore, for this case we change our robustness measure and use the CW attack (Carlini & Wagner, 2017) which will encounter fewer numerical issues because its loss function does not have a softmax component and does not saturate. Attacking the model from the natural source with CW-20 completely breaks the model and achieves 0.00% robustness. Most interestingly, attacking the model transferred from a robust source using the CW objective maintains robustness even when the number of hidden layers increases.

Figure 4: Training an MLP for CIFAR-100 on top of the robust feature extractors from ImageNet. The x-axis corresponds to the number of hidden layers (0 is a linear classifier and corresponds to experiments in section 4.1.1). Robustness stems from robust feature extractors. Adding more layers on top of this extractor does not hurt robustness. Interestingly, simply adding more layers does not improve the validation accuracy and just results in more overfitting (*i.e.* training accuracy becomes 100%). We can slightly improve generalization using batch norm (BN) and dropout (DO).

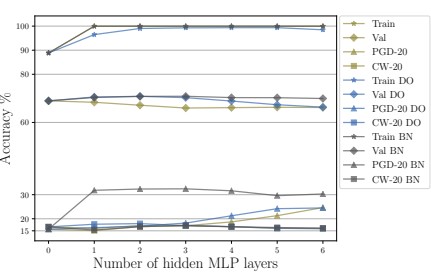

## 5 ANALYSIS: ROBUST FEATURE EXTRACTORS ARE FILTERS

Our experiments suggest that the robustness of neural networks arises in large part from the presence of robust feature extractors. We have used this observation to transfer both robustness and accuracy between domains using transfer-learning. However, we have not yet fully delved into what it means to have a robust feature extractor. Through visualizations, Tsipras et al. (2018) studied how adversarial training causes the image gradients of neural networks to exhibit meaningful generative behavior. In other words, adversarial perturbations on hardened networks "look like" the class into which the image is perturbed. Given that optimization-based attacks build adversarial examples using the image gradient, we also visualize the image gradients of our transferred models to see if they exhibit the same generative behavior as adversarially trained nets.

Fig. 5 plots the gradient of the loss w.r.t. the input image for models obtained by re-training only the last layer, and also for the case where we train MLPs on top of a robust feature extractor. The gradients for the transfer-learned models with a robust source are interpretable and "look like" the adversarial object class, while the gradients of models transferred from a natural source do not. This interpretatbility comes despite the fact that the source model was hardened against attacks on one dataset, and the transferred model is being tested on object classes from another. Also, we see that adding more layers on top of the feature extractor, which often leads to over-fitting, does not make gradients less interpretable. This latter observation is consistent with our observation that added layers preserve robustness(Fig. 4).

These observations, together with the success of robust transfer learning, leads us to speculate that a robust model's feature extractors act as a "filter" that ignores irrelevant parts of the image.

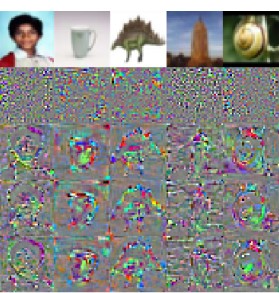

Figure 5: Gradients of the loss w.r.t to input images for the CIFAR-100 transfer learning experiments of sections 4.1.1 & 4.2.1. The top row contains sample CIFAR-100 images. Other rows contain image gradients of the model loss. The second row is for a model transferred from a naturally trained ImageNet source. Rows 3-5 are for models transferred from a robust ImageNet source. These rows correspond to an MLP with 0 (row 3), 1 (row 4), and 2 (row 5) hidden layers on top of the robust feature extractor. The gradients in the last three rows all show interpretable generative behavior.

## 6  END-TO-END TRAINING WITHOUT FORGETTING

As discussed in section 4, transfer learning can preserve robustness of the robust source model. However, it comes at the cost of decreased validation accuracy on natural examples compared to the case where we use a naturally trained source model. Consequently, there seems to be a trade-off between generalization and robustness based on the choice of the source model. For any given classifier, the trade-off between generalization and robustness is the subject of recent research (Tsipras et al., 2018; Zhang et al., 2019; Shafahi et al., 2019a). In this section, we intend to improve the overall performance of classifiers transferred from a robust source model by improving their generalization on natural images. To do so, unlike previous sections where we froze the feature extractor mainly to preserve robustness, we fine tune the feature extractor parameters $\theta$. Ideally, we should learn to perform well on the target dataset without catastrophically forgetting the robustness of the source model. To achieve this, we utilize lifelong learning methods.

Learning without Forgetting (LwF) (Li & Hoiem, 2018) is a method for overcoming catastrophic forgetting. The method is based on distillation. In this framework, we train the target model with a loss that includes a distillation term from the previous model.

$$\min_{w,\theta} \quad l(z(x,\theta),y,w) + \lambda_d \cdot d(z(x,\theta),z_0(x,\theta_r^*)) \tag{3}$$

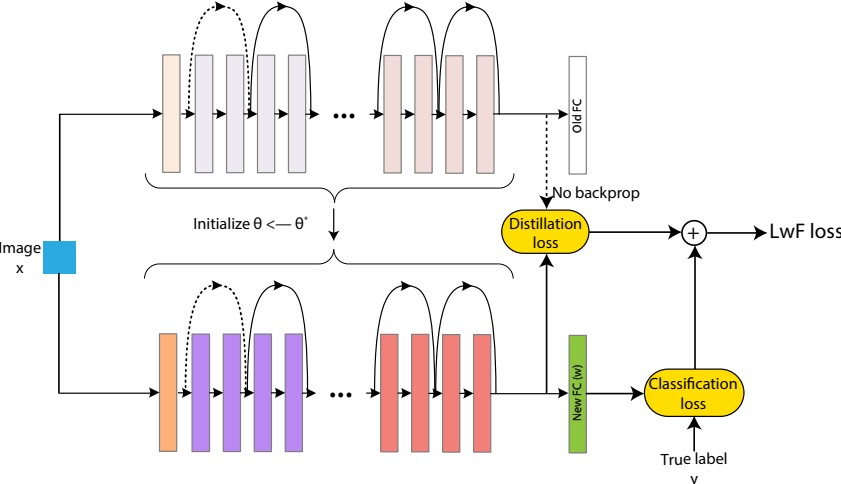

Figure 6: Our LwF loss has a term that enforces the similarity of feature representations (*i.e.* penultimate layer activations) between the source model and the fine-tuned model.

where, in our method, $\lambda_d$ is the feature representation similarity penalty, and $d$ is some distance metric between the robust model's feature representations $z_0(x,\theta_r^*)$ and the current model's feature representations $z(x,\theta)$. Unlike the original LwF paper that used a distilled loss from Hinton et al. (2015) and applies distillation to the logits, we simply choose $d$ to be the $\ell_2$-norm and apply

distillation to the penultimate layer[5]. Our loss is designed to make the feature representations of the source and target network similar, thus preserving the robust feature representations (Fig. 6). Ideally, $z(x, \theta) \approx z(x, \theta_r^*)$. To speed up training, given robust feature extractor parameters $\theta_r^*$, we store $z_0(x, \theta_r^*)$ for the images of the target task and load this from memory (*i.e.* offline) instead of performing a forward pass through the robust source network online. Therefore, in the experiments related to LwF, we do not train with data augmentation because we have not pre-computed $z(x_a, \theta_r^*)$, where $x_a$ is the augmented image. Empirically we verified that $d(z(x, \theta_r^*), z(x_a, \theta_r^*))$ was not negligible[6].

To improve performance, we follow a warm-start scheme and only train the fully connected parameters $w$ early in training. We then cut the learning rate and continue fine tuning both feature extractors ($\theta$) and $w$. In our experiments, we use a learning rate of 0.001, and the warm-start makes up half of the total training iterations. Starting from the pre-trained source model, we train for a total of 20,000 iterations with batch-size 128. The results with an adversarially trained CIFAR-100 model as source and CIFAR-10 as target are summarized in Table 4.[7]

As can be seen, having a LwF-type regularizer helps in maintaining robustness and also results in a considerable increase in validation accuracy. The trade-off between robustness and generalization can be controlled by the choice of $\lambda_d$. It seems that for some choices of $\lambda_d$ such as 0.1, robustness also increases. However, in hindsight, the increase in accuracy on PGD-20 adversarial examples is not solely due to improvement in robustness. It is due to the fact that the validation accuracy has increased and we have a better classifier overall. For easier comparisons, we have provided the transfer results without LwF at the bottom of Table 4. Note that using LwF, we can keep the robustness of the source model and also achieve clean validation accuracy comparable to a model that uses naturally trained feature extractors. In the supplementary, we show that similar conclusions can be drawn for the split CIFAR-100 task.

Table 4: Distilling robust features using learning without forgetting. The bottom rows show results from transfer learning with a frozen feature extractor. The '+' sign refers to using augmentation.

| Source $\rightarrow$ Target Dataset | Source Model | $\lambda_d$ | val. | PGD-20 ($\epsilon = 8$) |
|---|---|---|---|---|
| | | 1e-7 | 89.07% | 0.61% |
| | | 0.001 | 86.15% | 4.70% |
| CIFAR-100+ $\rightarrow$ CIFAR-10 | robust | 0.0025 | 81.90% | 15.42% |
| | | **0.005** | **79.35%** | **17.61%** |
| | | 0.01 | 77.73% | 17.55% |
| | | 0.1 | 73.39% | 18.62% |
| CIFAR-100+ $\rightarrow$ CIFAR-10+ | natural | NA | 83.05% | 0.00% |
| | robust | NA | 72.05% | 17.70% |

### DECREASING GENERALIZATION GAP OF ADVERSARIALLY TRAINED NETWORKS

We demonstrated in our transfer experiments that using our LwF-type loss, can help decrease the generalization gap while preserving robustness. In this section, we assume that the source domain is the adversarial example domain of a dataset and the target domain is the clean example domain of the same dataset. This experiment can be seen as applying transfer learning from the adversarial example domain to the natural example domain while preventing forgetting the adversarial domain.

In the case where the source and target datasets are the same (Transferring from a robust CIFAR-100 model to CIFAR-100), by applying our LwF-type loss, we can improve the generalization of robust models. Our results are summarized in Table 5.

---

[5]We do so since in section 3 we found the source of robustness to be the feature extractors and this observation was later reinforced due to the empirical results in section 4

[6]The analysis is in the supplementary.

[7]Source code for LwF-based experiments: https://github.com/ashafahi/RobustTransferLWF

Table 5: Decreasing generalization gap by transferring with LwF. For reference, last row shows results from adversarial training CIFAR-100. The '+' sign refers to using augmentation. ($\epsilon = 8$)

| Source $\rightarrow$ Target Dataset | Source Model | $\lambda_d$ | val. | PGD-20 | CW-20 |
|---|---|---|---|---|---|
| CIFAR-100+ $\rightarrow$ CIFAR-100 | robust | 1e-5 | 61.53% | 21.83% | 21.97% |
| | | **5e-5** | **61.71%** | **23.44%** | **22.94%** |
| | | 1e-4 | 61.38% | 23.95% | 23.40% |
| | | 0.001 | 60.17% | 24.31% | 23.17% |
| | | 0.01 | 59.87% | 24.10% | 22.96% |
| CIFAR-100+ | robust | NA | 59.87% | 22.76% | 23.16% |

## 7 CONCLUSION

We identified the feature extractors of adversarially trained models as a source of robustness, and use this observation to transfer robustness to new problems domains without adversarial training. While transferring from a natural model can achieve higher validation accuracy in comparison to transferring from a robust model, we can close the gap and maintain the initial transferred robustness by borrowing ideas from the lifelong learning literature. The success of this methods suggests that a robust feature extractor is effectively a filter that sifts out relevant components of an image that are needed to assign class labels. We hope that the insights from this study enable practitioners to build robust models in situations with limited labeled training data, or when the cost and complexity of adversarial training from scratch is untenable.

**Acknowledgements:** Goldstein and his students were supported by the DARPA QED for RML program, the DARPA GARD program, and the National Science Foundation.

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

## A  EXPERIMENT DETAILS

### A.1  LWF-BASED EXPERIMENTS

In our LWF-based experiments, we use a batch-size of 128, a fixed learning-rate of 1e-2m, and fine-tune for an additional 20,000 iterations. The first 10,000 iterations are used for warm-start; during which we only update the final fully connected layer's weights. During the remaining 10,000 iterations, we update all of the weights but do not update the batch-normalization parameters.

### A.2  IMAGENET TO CIFAR EXPERIMENTS

When freezing the feature extractor and fine-tuning on adversarial examples, we train the last fully connected layer's weights for 50 epochs using batch-size=128. We start with an initial learning rate of 0.01 and drop the learning rate to 0.001 at epoch 30. In the case of fine-tuning on adversarial examples, we generate the adversarial examples using a 3 step PGD attack with step-size 3 and a perturbation bound $\epsilon = 5$.

### A.3  FREE TRAINING EXPERIMENTS

In all of our free-training experiments where we train the u-ResNet-50, we train for 90 epochs using a batch-size of 128. The initial learning rate used is 0.1 and we drop it by a factor of 10 at epochs 30 and 60. We use a replay parameter $m = 4$ and perturbation bound $\epsilon = 5$.

## B  THE DISTANCE BETWEEN FEATURE REPRESENTATIONS OF NATURAL IMAGES AND AUGMENTED IMAGES

To speed up the LwF experiments, we did not use data augmentation during training. Instead of computing the robust feature representations on the fly, before starting training on the new target task, we passed the entire training data of the target task through the robust network and stored the feature representation vector. If we were doing data augmentation, we would have to pass the entire augmented training data through the network, which would be slow and memory intensive. Alternatively, we could use the robust feature representation of the non-augmented images instead. The latter would have been feasible if the distance between the robust feature representations of the non-augmented and augmented images were very small. However, as shown in fig 7, this quantity is not often negligible.

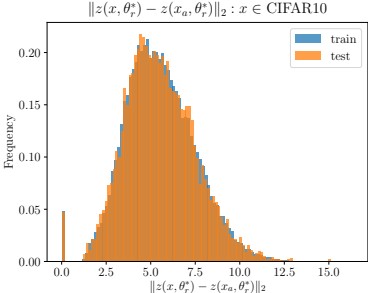
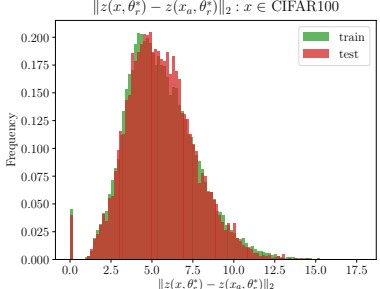

(a) Histogram of $\|z(x, \theta_r^*), z(x_a, \theta_r^*)\|_2$ for CIFAR-10 dataset, given $\theta_r^*$ for CIFAR-100 dataset. The mean for both training and test examples is $\simeq 5.53$

(b) Histogram of $\|z(x, \theta_r^*), z(x_a, \theta_r^*)\|_2$ for CIFAR-100 dataset, given $\theta_r^*$ for CIFAR-100 dataset. The mean for both training and test examples is $\simeq 5.57$

Figure 7: Figures 7a and 7b both show that the values of $\|z(x, \theta_r^*), z(x_a, \theta_r^*)\|_2$ are high most of the time, consequently, LwF is better done without data augmentation.

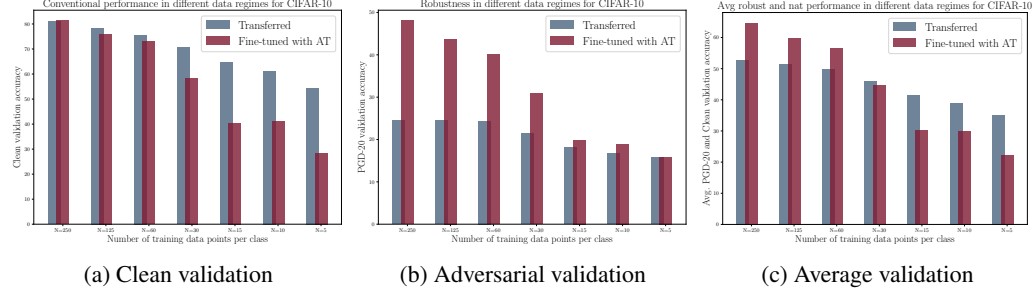

|     (a) Clean validation     |     (b) Adversarial validation     |     (c) Average validation     |

Figure 8: When the number of training data per-class is very limited (right bars), adversarially robust transfer learning [Transferred] is better overall. However, as the number of training data increases (left bars), fine-tuning with adversarial examples of the target domain [Fine-tuned with AT] results in an overall better performing model. Adversarially robust transfer learning is $3\times$ faster than fine-tuning with adversarial examples of the target domain.

## C    LOW-DATA REGIME TRANSFER LEARNING FROM IMAGENET TO CIFAR-10

In section 4.2, we illustrated that robust transfer learning is most beneficial where we have limited data (i.e., limited number of training data per class). We used CIFAR-100 as an illustrative target dataset. However, the overall results are not data set dependent. When we have limited number of training data, robust transfer learning results in the highest overall performance (see Fig. 8 for transferring from a robust ImageNet model to smaller instances of CIFAR-10).

## D    LwF-BASED ROBUST TRANSFER LEARNING FOR SIMILAR SOURCE AND TARGET DATASETS

In Table 6 we conduct LwF experiments on the split CIFAR-100 task which is more suited for transfer learning due to the similarities between the source and target datasets. In these situations, the LwF regularizer on the feature representations still works and can improve generalization without becoming vulnerable to adversarial examples. If we take the average performance of the robust classifiers on the split tasks (average of robust half CIFAR-100 and the LwF setting model for $\lambda_d = 0.01$) we get $(63.32 + 64.96)/2 = 64.14\%$ average validation accuracy and $20.42\%$ average robustness which is comparable with the case that we had adversarially trained the entire CIFAR-100 dataset (Table 1).

Table 6: Distilling robust features using LwF for the split CIFAR-100 task. For reference, we have included the results from transfer learning by freezing the features at the bottom of the table.

| Source → Target Dataset | Source Model | $\lambda_d$ | val. | PGD-20 |
|---|---|---|---|---|
| | | 0.001 | 73.30% | 1.92% |
| | | 0.005 | 66.96% | 10.52% |
| CIFAR-100+ (1/2) → CIFAR-100 (other 1/2) | robust | **0.01** | **63.32%** | **15.68%** |
| | | 0.1 | 55.14% | 17.26% |
| CIFAR-100+ (1/2) → CIFAR-100+ (other 1/2) | natural | NA | 71.44% | 0.00% |
| | robust | NA | 58.48% | 15.86% |

## E    IMPROVING GENERALIZATION OF THE CIFAR-10 ADVERSARIALLY TRAINED MODEL

Similar to the case of improving the generalization for CIFAR-100, we use our LwF-based loss function to transfer from the robust CIFAR-10 domain to the natural CIFAR-10 domain. We summarize the results in Table 7.

Table 7: Decreasing generalization gap by transferring with LwF. For reference, last row shows results from adversarial training CIFAR-10. The '+' sign refers to using augmentation. ($\epsilon = 8$)

| Source $\rightarrow$ Target Dataset | Source Model | $\lambda_d$ | val. | PGD-20 | CW-20 |
|---|---|---|---|---|---|
| CIFAR-10+ $\rightarrow$ CIFAR-10 | robust | 1e-5 | 88.16% | 45.31% | 46.15% |
|  |  | **5e-5** | **88.08%** | **46.24%** | **47.06%** |
|  |  | 1e-4 | 87.81% | 46.54% | 47.29% |
|  |  | 5e-4 | 87.44% | 46.36% | 47.16% |
|  |  | 0.001 | 87.31% | 46.27% | 47.06% |
|  |  | 0.01 | 87.27% | 46.09% | 46.92% |
|  |  | 0.1 | 87.49% | 46.11% | 46.84% |
| CIFAR-10+ | robust | NA | 87.25% | 45.84% | 46.96% |

