# OpenReview forum: "Adversarially robust transfer learning"
_ICLR.cc/2020/Conference — Accept (Poster)_

### Official Review · AnonReviewer1 · 2019-10-21
**Official Blind Review #1**

**Rating:** 8

**Review:**

Paper summary: This paper explores the problem of robustly transfer learning using only standard training (as opposed to adversarial training (AT)) on the target domain. The authors start by highlighting that intermediate representations learned by adversarially trained networks are themselves fairly robust. Then they propose two strategies for robust transfer from a robust model trained on the source domain: (1) naturally fine-tuning the final linear layer on the target domain and (2) naturally fine-tuning all the layers using lifelong learning strategies. They study transfer between CIFAR10 and CIFAR100, as well as, from ImageNet to CIFAR10/100.

High-level comments: Overall, I find the paper interesting and well-written. Prior work from Hendrycks et al. showed that AT on the source domain followed by *adversarial* fine-tuning on the target domain attains better performance as compared to just AT on the target domain. The main contribution of this paper is to show that using instead careful *natural* fine-tuning on the target domain is sufficient to recover a reasonable amount of this robustness.

Even though the clean/robust accuracy of the proposed approach is lower than just doing adversarial training/prior work from Hendrycks et al., I feel this paper could be useful to the community for two main reasons:

1. The authors perform a nice exploration of the thesis that robust models have robust representations, and how this connects to transfer learning. In particular, the experiments in Figure 1 (the effect of naturally re-training later layers of a robust network on its robustness) and Figure 5 (where the authors show that their naturally fine-tuned models have some of the unexpected benefits from Tsipras et al.) seem particularly interesting.

2. Despite its lower accuracy, this approach could be useful in settings where data is scarce or compute is expensive, and hence adversarial training on the target domain is not successful.

Specific comments/questions:

i. Could the authors clarify what they mean by point 3 (re: validation accuracy drop) below Table 3? As far as I can tell, the drop in clean validation accuracy between Table 1 and Table 2 are similar for both the naturally and adversarially pre-trained models.

ii. In Table 2, it would also be interesting to see the performance when the source domain is CIFAR10 and the target domain is CIFAR100.

iii. For Table 2 is the eps=8? This should be mentioned in the caption as it is important to highlight that Tables 2 and 3 are not directly comparable.

iv. Why isn’t experiment in Figure 3 should be repeated for CIFAR10 as well? The authors should add this result to the paper, even if in the appendix. It is important to verify that this trend is not specific to CIFAR100 and holds across datasets (even though CIFAR10/100 are not too different).

v. The comment at the end of page 6 re: natural model is confusing (“Note, this seems to...perfectly”)---as far as I can tell, Figure 4 does not include the results of fine-tuning a naturally trained model.

vi. General comment motivated by the comment ("Note...perfectly") mentioned 5 above: For all the adversarial evaluation in the paper, the authors should also try CW attacks/black-box attacks to get a more confident estimate of their model’s robustness.

vii. The authors reference prior work on the tradeoff between robustness and accuracy and motivate Section 6 as an avenue to alleviate this trade-off for their model. However, I don’t see the lower performance of their model as an instance of this trade-off---the model in the paper performs worse in terms of both clean and adversarial accuracy. The approach proposed in Section 6 seems interesting, but more as an approach to improve the *overall* performance of the model. The authors mention this in retrospect, but I think the narrative of this section should be modified to make this clearer.

viii. In Table 5, in the experiments corresponding to CIFAR100+ -> CIFAR100, is the dataset split into two halves (for the source and target domains) or is the fine-tuning performed on the same data. In general, I find it odd that natural fine-tuning on the *same data* can improve both the clean and adversarial accuracy of the model (compared to the CIFAR100+ robust baseline). Is the robust model trained long enough/with enough hyperparameter search?

Overall, the exploration in the paper seems novel and could be useful to the community. Thus, I recommend acceptance.


**Experience Assessment:**

I have published in this field for several years.

**Review Assessment: Checking Correctness Of Derivations And Theory:**

N/A

**Review Assessment: Checking Correctness Of Experiments:**

I carefully checked the experiments.

**Review Assessment: Thoroughness In Paper Reading:**

I read the paper thoroughly.

---

> ### Author Response · Authors · 2019-11-14
> **Thanks for your feedback [1/2]**
>
> Thank you very much for your encouraging review. Below we have tried to address all of your specific comments/questions.
>
> > "i. Could the authors clarify what they mean by point 3 (re: validation accuracy drop) below Table 3? As far as I can tell, the drop in clean validation accuracy between Table 1 and Table 2 are similar for both the naturally and adversarially pre-trained models."
>
> The reviewer is correct! We can see the ambiguity. We meant to say that the clean validation accuracy is less for the case where we use a robust source model compared to the case where we use a conventionally trained source model. In terms of the drop in clean validation accuracy ((80.20-71.44=8.76) vs (64.96-58.48=6.48)), the results are similar. Thank you for pointing this out. We have clarified this in the revision.
>
> > "ii. In Table 2, it would also be interesting to see the performance when the source domain is CIFAR10 and the target domain is CIFAR100."
>
> We have added the requested experiment to Table 2. Using CIFAR10 as the source domain and transferring to CIFAR100 is not as good as using ImageNet as the source domain. We hypothesize that this is because the robust features required for the 10-class classification problem is not representative enough for the 100-class fine-grained classes of CIFAR-100. Note that we cannot even perfectly fit the training data and we only get about 50% accuracy on the CIFAR-100 training data. This is opposed to >80% training accuracy when we use ImageNet as the source domain.
>
> > "iii. For Table 2 is the eps=8? This should be mentioned in the caption as it is important to highlight that Tables 2 and 3 are not directly comparable."
>
> Yes. For Tables 1 and 2, we use eps=8 since for CIFAR models, it is common to train models against this epsilon value. However, for Table 3, we use eps=5 since the source model is an ImageNet model and (Free) adversarially training ImageNet models to resist non-targeted attacks with eps=8 is not feasible since eps=8 is simply a large perturbation for non-targeted attacks on ImageNet which has 1000 classes. Per your suggestion, we have included the epsilons in the captions for all tables.
>
> "iv. Why isn’t experiment in Figure 3 repeated for CIFAR10 as well? The authors should add this result to the paper, even if in the appendix. It is important to verify that this trend is not specific to CIFAR100 and holds across datasets (even though CIFAR10/100 are not too different)."
>
> We focused on CIFAR-100 since it is a task where by default the number of training data per-class is limited. However, similar trends can be seen for CIFAR-10 whenever we move in the very sparse training data points per-class regime. In the low-data regime, transfer learning by using natural examples results in an overall better classifier than fine-tuning on adversarial examples. Per your suggestion, we have included the results in the supplementary materials of the revision.
>
> "v. The comment at the end of page 6 re: natural model is confusing (“Note, this seems to...perfectly”)---as far as I can tell, Figure 4 does not include the results of fine-tuning a naturally trained model."
>
> You are again correct that Figure 4 does not include the results for using a naturally trained model as the source. However, that sentence is based on a fast experiment which we ran but did not include in Figure 4. We have tried to clarify this in the revision by adding the quantitative detailed results. Specifically, we have stated: “We speculate that the increase in robustness against PGD attacks by increasing the number of layers of the MLP may be due to vanishing gradients. We verified this by observing that as we increase the number of hidden layers of the MLP, robustness increases, even when we use a pre-trained naturally trained model. While for the case where we have no hidden layers, robustness is 0 on CIFAR100, if our MLP has 1, 2, 3, or 5 hidden layers, our robustness against PGD attacks would be 0.03%, 0.09%, 0.31% and 6.61%, respectively.”

---

> ### Author Response · Authors · 2019-11-14
> **Thanks for your feedback [2/2]**
>
> >"vi. General comment motivated by the comment ("Note...perfectly") mentioned 5 above: For all the adversarial evaluation in the paper, the authors should also try CW attacks/black-box attacks to get a more confident estimate of their model’s robustness."
>
> Adding results on attacking our robust models using different attack types is definitely a good idea and we have done that in the revised document. Overall, the results are consistent with before and the robustness against CW and PGD attacks are often within 1% of each other.
> During the rebuttal period, we have also evaluated our ImageNet->CIFAR100 models on adversarial examples made with NAttack (a black-box attack [1]). This black-box attack (which is technically a gradient-free attack) is very expensive. NAttack takes about 5 days to evaluate all 10000 CIFAR-100 test images on a single 2080Ti.
> NAttack is a gradient-free attack based on constrained Natural Evolution Strategy. Its requires access to the logits of the model, which is a relatively strong assumption compared with other black-box settings such as [2]. Nevertheless for the first 5363 images (we wanted to post this soon to enable further discussion but will update the final number when the experiment is over), with 500 iterations and a population size of 300, our ImageNet->CIFAR100 model is able to maintain an accuracy of 17.15% against black-box attacks (as expected, the black-box attack is weaker by about 2% than the white-box PGD-20 attack).
>
> > "vii. The authors reference prior work on the tradeoff between robustness and accuracy and motivate Section 6 as an avenue to alleviate this trade-off for their model. However, I don’t see the lower performance of their model as an instance of this trade-off---the model in the paper performs worse in terms of both clean and adversarial accuracy. The approach proposed in Section 6 seems interesting, but more as an approach to improve the *overall* performance of the model. The authors mention this in retrospect, but I think the narrative of this section should be modified to make this clearer."
>
> We agree that section 6 is improving the overall performance of the model. The LWF parameter $\lambda$ controls how much weight we put on robustness vs generalization. While $\lambda$ does control the trade-off, as the reviewer has stated, the overall section is designed to improve the generalization and hence the overall performance of the target model. We have clarified this in the revision.
>
> > "viii. In Table 5, in the experiments corresponding to CIFAR100+ -> CIFAR100, is the dataset split into two halves (for the source and target domains) or is the fine-tuning performed on the same data. In general, I find it odd that natural fine-tuning on the *same data* can improve both the clean and adversarial accuracy of the model (compared to the CIFAR100+ robust baseline). Is the robust model trained long enough/with enough hyperparameter search?"
>
> This is done on the entire dataset (same data for source and target). We use the standard hyper-parameters used for training a WRN32-10 by Madry et al [3] for training the robust model and is trained long enough (i.e., until it reaches 100% training accuracy on training data adversarial examples). Note that we use the robust CIFAR100+ model as the source. In section 6, we focus on improving the generalization on natural data. However, in some cases, improving generalization also increases robustness (especially with the LwF loss). This seemingly odd phenomenon is not limited to the case where we use the same dataset and it can also be seen in the other results table for section 6 (Table 4). We try to address this interesting observation in section 6 by saying that “ The increase in accuracy on PGD-20 adversarial examples is not solely due to improvements in robustness. It is also due to the fact that the validation accuracy has increased and we have a better classifier overall.”. In the appendix, we have also included a similar analysis by using the officially released pre-trained robust CIFAR-10 model from Madry et al. as a source. We can see a similar trend in that case as well.
>
> [1] Li, Yandong, et al. "NATTACK: Learning the Distributions of Adversarial Examples for an Improved Black-Box Attack on Deep Neural Networks." International Conference on Machine Learning. 2019.
> [2] Papernot, Nicolas, et al. "Practical black-box attacks against machine learning." Proceedings of the 2017 ACM on Asia conference on computer and communications security. ACM, 2017.
> [3] https://github.com/MadryLab/cifar10_challenge

---

> > ### Comment · AnonReviewer1 · 2019-11-15
> > **Re: Author Response**
> >
> > Thank you for addressing my questions and remarks. Based on these modifications, I have updated my score by one point.
> >
> > Minor comment: In the paper revision, the order in which Table  3 and Table 2 appear in the paper is reversed. The authors should update this in the final manuscript.

---

### Official Review · AnonReviewer2 · 2019-10-23
**Official Blind Review #2**

**Rating:** 8

**Review:**

The paper studies transfer learning from the point of view of adversarial robustness. The goal is, given a robust deep neural network classifier for a source domain, learn a robust classifier for a target domain as efficiently and with as few samples as possible. The authors empirically evaluate different strategies and compare with relevant baselines.

At a high level, the paper addresses an interesting problem. Robust models are quite computationally and sample intensive to train, so exploring pre-training is a reasonable way to deal with small datasets or computational constraints.

The authors perform a diverse set of experiments from which I identified the following individual contributions:

a) Retraining the last layer of the model on natural examples preserves robustness. Robustness degrades smoothly when pre-training progressively more layers.
This is an interesting contribution providing evidence that robust models do learn in fact robust input representations/features.

b) Transferring a learned robust representation from a source to a target domain preserves its robustness (training a linear layer on top of it leads to a robust classifier).
This provides further evidence that robust models learn _general purpose_ robust features of the input, while establishing robust pre-training as a valid strategy for cheaper robust models. The baselines considered are: 1) adversarial training on target domain which always underperforms the proposed method, 2) fine-tuning on adversarial samples from the target domain which performs better when there are a lot of samples from that domain and worse when there are only a few (this method is also more computationally expensive than transfer learning).

c) The "perceptually-aligned" saliency maps of Tsipras et al. 2018 are also a property of robust models obtained through transfer learning.
This illustrates that these saliency maps can also arise for out-of-distribution inputs and hence are likely to correspond to general, high-level features.

d) Fine-tuning all layers of the model while ensuring that the representations stay close to the original ones for natural examples can lead to transfer with improved validation accuracy and robustness (even when the source and target domains are the same).
This is an interesting improvement over the simpler transfer methods producing competitive results.

Overall, the paper contains an experimental study that, in my opinion, is thorough, presents interesting findings, and contains the necessary ablations. I believe that this paper would be of interest to the adversarial ML community and I hence recommend acceptance.


**Experience Assessment:**

I have published in this field for several years.

**Review Assessment: Checking Correctness Of Derivations And Theory:**

I carefully checked the derivations and theory.

**Review Assessment: Checking Correctness Of Experiments:**

I carefully checked the experiments.

**Review Assessment: Thoroughness In Paper Reading:**

I read the paper thoroughly.

---

> ### Author Response · Authors · 2019-11-14
> **Thanks for your feedback**
>
> Thank you very much for your encouraging review. We indeed hope that this work encourages the practitioners which are interested in having robust models but have limited data or resources to use adversarially robust pre-trained networks.

---

### Official Review · AnonReviewer3 · 2019-10-23
**Official Blind Review #3**

**Rating:** 1

**Review:**

Summary
-------
This paper addresses the problem of performing robust transfer learning. A first contribution of the paper is to robust and classic training with respect to usual validation accuracy and robustness to adversarial attacks on the CIFAR task. Then, the same comparison is made on a transfer learning task. The transfer learning setting is then completed by studying transfer from ImageNet-based models with a particular attention to low-data regime and training deeper networks on top of the feature extractor. An analysis of robust features is provided and finally the authors studies the interest of  Learning without Forgetting strategies to provide robust transfer. The tendency s to obtain the Best performance from robust-trained source models having a good validation accuracy.


Overall
------
The paper presents  a study of robust transfer learning that can be interesting for practitioners to know the type of results that can be obtained by robust transfer learning. However, I feel that the results obtained are rather expected and the paper does not provide some interesting methodological contribution that could help to develop robust transfer training.

Comments
---------

The results obtained in Section 3, 4 and 6 are rather expected and similar.  I think that the paper could benefit by reducing these 3 sections in only one section where the results obtained can be summarized in one big table and two or three figures for example - the complete set of results can then be reported in the supplementary section.

Then, if the contribution of the paper is to propose to focus on robust transfer learning including a Learning without Forgetting strategy, the authors should then focus more on this part and analyze better the behavior of learning.
In particular, the combination between distillation and robust training is certainly interesting, and trying to propose a methodological framework for doing robust training in this context would certainly result in a more significant contribution. How to constrain the feature extraction layers, how to make use of them with distillation and additionally what are the additional contraints/additions that can be made to learning problem (3) to improve robust transfer are some important questions.

So far, the contribution appears to me rather limited for ICLR. If we restrict to the part related to experimental comparisons made, they are restricted to particular trainings and datasets with specific PGD attacks. The contribution would have been stronger is different types of adversarial attacks with different parameters have been studied and analyzed.

**Experience Assessment:**

I have read many papers in this area.

**Review Assessment: Checking Correctness Of Derivations And Theory:**

I assessed the sensibility of the derivations and theory.

**Review Assessment: Checking Correctness Of Experiments:**

I assessed the sensibility of the experiments.

**Review Assessment: Thoroughness In Paper Reading:**

I read the paper at least twice and used my best judgement in assessing the paper.

---

> ### Author Response · Authors · 2019-11-14
> **Thanks for your feedback [1/2]**
>
> Thank you for the comments and for taking the time to review our work. In what follows we have provided a detailed response to your concerns:
> > “I feel that the results obtained are rather expected and the paper does not provide some interesting methodological contribution that could help to develop robust transfer training. “
>
> Our contributions are two-fold:  First, we observe that adversarially trained networks contain generic robust features extractors that maintain their robustness regardless of the downstream task.  Perhaps this is not a “surprise,” but we don’t think this observation is obvious or trivial either; it is hypothetically possible that the feature extractors strongly adapt themselves to the classifier head rather than learning the generic behavior that we observed.
>
> Our second contribution is a detailed empirical study of how to maximize the transfer of robustness from one classifier to another so that practitioners can maximize the benefits of the observation above.  For this purpose, we explore the following issues...
> 1) In conventional transfer learning, it is common to retrain multiple layers of a network. However, our experiments illustrate robustness transfers best using just one layer (See Fig. 1).
> 2) In conventional transfer learning, one fine-tunes a network on data from the target domain. When trying to transfer robustness, it is most natural to fine-tune on adversarial examples in the target domain. Interestingly (and unintuitively), the best overall performance (on both clean and adversarial images) in the low-data regime  is often achieved by fine-tuning *only* on clean/natural examples (See Fig. 3). Fine tuning on adversarial examples only pays off in the large data regime (note, however, that transfer learning often takes place in the low data regime).
> 3) In the simpler (and more expected case) where we only retrain the classification layer on the target task, we show visualizations to uncover why using a robust source model (feature extractor) results in some robustness (See Fig. 5).
> 4) We show that we can improve the overall performance of a model and improve generalization on clean examples by fine-tuning all layers and at the same time using a LwF penalty to prevent the network from forgetting how to be robust. Note that our identification and discussion of sources of robustness motivates us to add a LwF-type loss for the *penultimate layer* as opposed to the logit layer (note the latter is more common for natural distillation and LwF).
>
> > “The results obtained in Section 3, 4 and 6 are rather expected and similar.  I think that the paper could benefit by reducing these 3 sections in only one section where the results obtained can be summarized in one big table and two or three figures for example - the complete set of results can then be reported in the supplementary section.”
>
> Thank you for the comment. We had separated each section since we believed every section has its own independent story and result. Section 3 is an exploratory section which focuses on finding the source of robustness. The experiments done here are isolated experiments where we use the same dataset, architectures, and hyper-parameters for the source and target models. Results of section 3 (i.e, the source of robustness is the feature extractor), justify using robust feature extractors for transfer learning. Section 4, is an experimental section which focuses on the traditional case of transfer learning (i.e., freezing the robust feature extractor) and shows that the results from section 3 also hold when the source and target datasets significantly differ. Section 5 is an explanatory section that explains why is it that we experimentally found that robustness transfers in section 4. Finally, section 6 deviates from the standard transfer learning case and focuses on retraining all layers without catastrophically forgetting the robust feature representations learned from the source domain. In section 6, we also show an interesting application of using an LwF-type loss for improving the overall performance of a robust model by using adversarial examples as the source task and natural examples as the target task.

---

> ### Author Response · Authors · 2019-11-14
> **Thanks for your feedback [2/2]**
>
> Remaining concerns:
> > “the combination between distillation and robust training is certainly interesting, and trying to propose a methodological framework for doing robust training in this context would certainly result in a more significant contribution. How to constrain the feature extraction layers, how to make use of them with distillation and additionally what are the additional contraints/additions that can be made to learning problem (3) to improve robust transfer are some important questions. “
>
> We agree that considering the combination between distillation and robust training is certainly interesting. Section 6, by itself is trying to tackle such a problem. However, our paper focuses on the task of robust transfer learning and one of our contributions is to propose using such distillation techniques for improving the overall performance of models trained with robust transfer learning. We make modifications to the standard distillation loss for our purpose as stated in Section 6. A more comprehensive study of using distillation for robust model compression or preventing catastrophic forgetting during distillation is certainly an interesting research direction.  However, we feel that this is outside the scope of this article, as it would prevent us from thoroughly exploring the full spectrum of transfer learning methods.
>
> > "If we restrict to the part related to experimental comparisons made, they are restricted to particular trainings and datasets with specific PGD attacks. The contribution would have been stronger is different types of adversarial attacks with different parameters have been studied and analyzed. “
>
> We think this comment is very constructive, and after discussion among the authors we agree that considering different types of attacks can further strengthen our paper. In the initial submission, we had considered the CW attack only for the case that we were concerned with gradient vanishing when we were training an MLP on top of the frozen source model’s robust feature extractor. In the revision, we have added results of attacking the models using different well-known attacks (including CW). See the updated tables.

---

### Decision · Program_Chairs · 2019-12-19

**Decision:**

Accept (Poster)

**Comment:**

This paper presents an empirical study towards understanding the transferability of robustness (of a deep model against adversarial examples) in the process of transfer learning across different tasks.

The paper received divergent reviews, and an in-depth discussion was raised among the reviewers.

+ Reviewers generally agree that the paper makes an interesting study to the robust ML community. The paper provides a nice exploration of the hypothesis that robust models learn robust intermediate representations, and leverages this insight to help in transferring robustness without adversarial training on every new target domain.

- Reviewers also have concerns that, as an experimental paper, it should perform a larger study on different datasets and transfer problems to eliminate the bias to specific tasks, and explore the behavior when the task relatedness increases or decreases.

AC agrees with the reviewers and encourages the authors to incorporate these constructive suggestions in the revision, in particular, explore more tasks with different task relatedness.

I recommend acceptance, assuming the comments will be fully addressed.